# HNF1A POU Domain Mutations Found in Japanese Liver Cancer Patients Cause Downregulation of *HNF4A* Promoter Activity with Possible Disruption in Transcription Networks

**DOI:** 10.3390/genes13030413

**Published:** 2022-02-24

**Authors:** Effi Haque, Aamir Salam Teeli, Dawid Winiarczyk, Masahiko Taguchi, Shun Sakuraba, Hidetoshi Kono, Paweł Leszczyński, Mariusz Pierzchała, Hiroaki Taniguchi

**Affiliations:** 1Institute of Genetics and Animal Biotechnology of the Polish Academy of Sciences, 05-552 Jastrzebiec, Poland; e.haque@igbzpan.pl (E.H.); teeliaamir7@gmail.com (A.S.T.); d.winiarczyk@igbzpan.pl (D.W.); p.leszczynski@igbzpan.pl (P.L.); m.pierzchala@igbzpan.pl (M.P.); 2Molecular Modeling and Simulation Group, National Institutes for Quantum Science and Technology, Kizugawa 619-0215, Japan; taguchi.masahiko@qst.go.jp (M.T.); sakuraba.shun@qst.go.jp (S.S.); kono.hidetoshi@qst.go.jp (H.K.)

**Keywords:** hepatocellular carcinoma, mutation, HNF1A, POU domain, HNF4A

## Abstract

Hepatocyte nuclear factor 1A (HNF1A) is the master regulator of liver homeostasis and organogenesis and regulates many aspects of hepatocyte functions. It acts as a tumor suppressor in the liver, evidenced by the increased proliferation in HNF1A knockout (KO) hepatocytes. Hence, we postulated that any loss-of-function variation in the gene structure or composition (mutation) could trigger dysfunction, including disrupted transcriptional networks in liver cells. From the International Cancer Genome Consortium (ICGC) database of cancer genomes, we identified several *HNF1A* mutations located in the functional Pit-Oct-Unc (POU) domain. In our biochemical analysis, we found that the *HNF1A* POU-domain mutations Y122C, R229Q and V259F suppressed *HNF4A* promoter activity and disrupted the binding of HNF1A to its target *HNF4A* promoter without any effect on the nuclear localization. Our results suggest that the decreased transcriptional activity of HNF1A mutants is due to impaired DNA binding. Through structural simulation analysis, we found that a V259F mutation was likely to affect DNA interaction by inducing large conformational changes in the N-terminal region of HNF1A. The results suggest that POU-domain mutations of HNF1A downregulate *HNF4A* gene expression. Therefore, to mimic the HNF1A mutation phenotype in transcription networks, we performed siRNA-mediated knockdown (KD) of HNF4A. Through RNA-Seq data analysis for the HNF4A KD, we found 748 differentially expressed genes (DEGs), of which 311 genes were downregulated (e.g., *HNF1A*, *ApoB* and *SOAT2*) and 437 genes were upregulated. Kyoto Encyclopedia of Genes and Genomes (KEGG) mapping revealed that the DEGs were involved in several signaling pathways (e.g., lipid and cholesterol metabolic pathways). Protein–protein network analysis suggested that the downregulated genes were related to lipid and cholesterol metabolism pathways, which are implicated in hepatocellular carcinoma (HCC) development. Our study demonstrates that mutations of HNF1A in the POU domain result in the downregulation of HNF1A target genes, including *HNF4A,* and this may trigger HCC development through the disruption of HNF4A–HNF1A transcriptional networks.

## 1. Introduction

Liver cancer is a major contributor to the cancer burden and one of the leading causes of cancer-dependent deaths worldwide [1,2]. The common risk factors for liver cancer development include alcohol consumption, hepatitis B and C virus infection, and metabolic diseases [3,4]. Most of these factors lead to genetic aberrations in hepatocytes, leading to their oncogenic transformation [3,4]. Researchers studying cancer cell genomes have undertaken several projects elucidating the genomic alterations present in different cancers, including liver cancer [5,6,7]. Next-generation sequencing (NGS) analysis data for liver cancer patients in which the most significantly mutated genes were *TP53*, *CTNNB1*, and *TERT* have been reported [7].

Besides these major driver gene mutations, other driver and tumor suppressor genes with well-established roles in liver function have been found. Among many gene mutations found through NGS-based mutational detection, hepatocyte nuclear factor 1A (*HNF1A*) was found to be a frequently mutated gene, one of the top 20 mutated genes in hepatocellular carcinoma (HCC) reported in the International Cancer Genome Consortium (ICGC) database (https://dcc.icgc.org/, accessed on 12 October 2020). HNF1A, a liver-enriched transcription factor, is present in embryonic tissues and plays a pivotal role in cellular differentiation and organ development [8]. *HNF1A* acts synergistically with *HNF4A* to regulate gene expression in various tissues, including the intestine and kidney [9,10]. In addition to its function in liver development, a recent study demonstrated that HNF1A knockout (KO) mice developed HCC due to fatty liver [11]. Moreover, in the HCC microenvironment, HNF1A inhibits Wingless-related integration site (Wnt) and nuclear factor kappa-B (NF-κB) signaling during metastasis and hepatocarcinogenesis [12,13]. On the other hand, the overexpression of HNF1A suppressed the proliferation of HCC and induced the expression of liver-specific genes in HCC cells, which caused cell cycle arrest [14]. These results support the idea that the dysfunction of HNF1A may cause hepatocarcinogenesis and HCC progression. However, while the role of HNF1A in different cancers has recently been examined [15,16,17], only a few studies have demonstrated a critical link between HNF1A mutations and the development of liver cancer.

Structurally, HNF1A has three domains: a dimerization domain, a DNA-binding domain, and a transactivation domain. The central DNA-binding domain is composed of a Pit-Oct-Unc (POU) homeodomain (POUh) and POU-specific (POUs) domain and is indispensable for efficient transcriptional activity [18]. HNF1A interacts with target DNA as a homodimer or heterodimer with HNF1B to regulate glucose metabolism, lipid metabolism, and detoxification [19,20,21]. HNF1A occupies the *HNF4A* promoter region and upregulates its expression as positive feedback [22]. Accordingly, a reduction in HNF4A has been associated with the reduced expression of HNF1A in young mice [23]. Similarly, HNF4A and HNF1A, together, form a network that regulates the expression of each as well as multiple liver-specific genes [22,24,25]. Additionally, our group reported for the first time that HNF4A G79C, F83C, and M125I mutations are loss-of-function mutations found in liver cancer patients, leading to a reduction in *HNF1A* gene expression and concomitantly, an increased risk of HCC development [26]. Several studies have demonstrated that HNF1A and HNF4A reciprocally regulate each other’s expression through DNA-binding-dependent and independent (protein–protein interaction) mechanisms [27,28]. These findings suggest that both HNF1A and HNF4A are critical regulators of liver function, and their dysfunction leads to liver cancer development. However, unlike for HNF4A mutations, the effects of HNF1A mutations on *HNF4A* gene regulation and HCC development remain elusive. Notably, much like HNF4A mutations, the ICGC and The Cancer Genome Atlas (TCGA) have reported mutations in the DNA-binding domain of HNF1A [29,30]. Previous studies have established that HNF1A mutations are associated with hepatocellular adenomas and maturity-onset diabetes of the young type 3 (MODY3) [30,31]. P112L and Q466X mutations of HNF1A have been associated with MODY [31]. Although mutations of HNF1A Q511L, E32*, and L214Q have also been identified in HCC [30,32,33], the effect of HNF1A POU domain mutation on the regulation of HNF4A and its downstream molecular mechanism to trigger HCC remain unknown.

In our study, we demonstrated that somatic mutations of HNF1A located in the POU domain are possible pathogenic mutations for hepatocarcinogenesis due to their disruption of *HNF4A* gene transcription. The mutations interfere with the ability of HNF1A to bind to the DNA of its target *HNF4A* promoter, and reduced transcriptional activity is observed. Moreover, structural analysis of the HNF1A V259F mutation revealed that it causes large conformational changes in the N-terminal region. However, RNA-Seq data for HNF4A siRNA knockdown (KD) in human hepatoma cell line (Huh7) cells suggested that the HNF4A mediated decrease in the expression of *HNF1A* and other genes is related to binding activity, the lipid and cholesterol metabolism pathways. These results suggest that proper transcriptional control between HNF1A and HNF4A maintains liver homeostasis and that the disruption of HNF1A–HNF4A transcriptional networks by mutations, aberrant expression or both may play a role in liver cancer development.

## 2. Materials and Methods

### 2.1. Cell Culture

Human embryonic kidney cells (HEK293:ATCC CRL-1573) and Huh7 cells were cultured in Dulbecco’s modified Eagle’s medium (DMEM) supplemented with 4.5 g/liter of glucose (Lonza, Basel, Switzerland, 10% fetal bovine serum (FBS) (EURx, Gdansk, Poland), 100 units/mL of penicillin, and 100 units/mL of streptomycin (Lonza). The cells were cultured under humidified conditions in an incubator at 5% CO_2_ and 37 °C.

### 2.2. Plasmids and Primers

To amplify the HNF1A sequence, we isolated the genomic DNA from non-immunogenic mouse hepatoma cells (Hepa1–6) using a Genomic Mini kit (A&A Biotechnology, Gdynia, Poland). The primers for the selected gene were designed based on the sequence located on the chromosome. Restriction sites were incorporated into the forward and reverse primers, respectively. The primers used for cloning the HNF1A plasmid are listed in Appendix A. A human HNF1A wild-type (WT) plasmid construct used in this study was procured from Addgene (Teddington, UK). Mutant variants of human HNF1A Y122C and V259F were created through site-directed mutagenesis by using a site-directed mutagenesis kit (Agilent Technologies, Santa Clara, CA, USA). The *HNF4A* P1 (−985 to +1 of the P1 *HNF4A* promoter) promoter was cloned into a basic pGL3 vector containing the luciferase gene (Promega, Madison, WI, USA) digested with KpnI and HindIII enzymes (Thermo Fisher Scientific, Waltham, MA, USA) using an In-Fusion® HD Cloning Kit (Takara, Shiga, Japan). The reporter constructs P2 (−371 to −37 from the HNF4A transcription start site) and P2-2200 (−2200 to −1 of the P2 *HNF4A* promoter) were purchased from Addgene. CMYC and FLAG CMV vectors were used for control experiments. Specific primers were designed for mutagenesis using the QuikChange Primer Design tool (Agilent Technologies). The Y122C, R229Q and V259F mutated sequences were confirmed using Sanger sequencing (Genomed, Warsaw, Poland). The primers used for the mutagenesis were also purchased from Genomed and are listed in Appendix A.

### 2.3. Reporter Assay

For the reporter assay, 5 × 10^4^ HEK293 cells and Huh7 cells were seeded in 24-well plates. After 24 h, the cells were transfected with 100 ng of the mouse and human plasmids indicated in the figures, using Lipofectamine 3000 (Thermo Fisher Scientific). The cells were transiently co-transfected with 500 ng of an *HNF4A* promoter–reporter construct containing consensus binding sites upstream of the firefly luciferase and 100 ng of a thymidine kinase promoter-Renilla luciferase reporter plasmid, as an internal control, using Lipofectamine 3000 (Thermo Fisher Scientific), according to the manufacturer’s instructions. Following 48 h of transfection, the cells were lysed and the luciferase activity was measured with a Luciferase Assay Kit (Promega), according to the enclosed protocol, using a Synergy LX luminometer (Biotek, Winooski, VT, USA).

### 2.4. Western Blotting

A total of 5 × 10^5^ HEK293 cells were plated in 6-well plates and transfected for overexpression with different HNF1A plasmids in amounts of 2 µg for 48 h, using Lipofectamine 3000 (Thermo Fisher Scientific), according to the manufacturer’s instructions. The nuclear protein concentrations from the HNF1A WT and HNF1A mutant cells were determined using the Pierce BCA Protein Assay Kit (Thermo Fisher Scientific, Waltham, MA, USA). The molecular weight of the protein was estimated with Precision Plus Protein WesternC Standards (Bio-Rad, Hercules, CA, USA). A total of 10 µg of each protein sample was loaded on an SDS-polyacrylamide gel (4% stacking gel; 12% resolving gel), separated, and transferred to a PVDF membrane (Merck Millipore, Burlington, MA, USA) by wet transfer. The membranes were blocked with 5% skim milk and then incubated with the antibodies. The blot was incubated overnight with mouse monoclonal Anti-Flag antibody (1:5000, Sigma) in 1% skim milk and 0.1% PBST at 4 °C, followed by incubation with HRP-conjugated anti-mouse IgG produced in goats (1:5000, Sigma-Aldrich, Saint Louis, MO, USA) in 1% skim milk and 0.1% PBST for 1 h at room temperature. For the siRNA KD experiment, we used rabbit monoclonal anti-HNF4A (1:1000, Cell Signaling Technology, Danvers, MA, USA) antibody and anti-rabbit IgG produced in goats (1:5000, Sigma-Aldrich). Anti-β-actin (1:1000, Cell Signaling Technology) was used as a loading control. The proteins were visualized using an ECL Western Blotting Analysis System (Amersham, Illinois, CA, USA) and ChemiDoc XRS + System (Bio-Rad, Hercules, CA, USA).

### 2.5. Immunofluorescence (IFC)

For IFC staining, 5 × 10^5^ HEK293 cells were plated in 6-well plates and transfected for overexpression with HNF1A WT and mutant plasmids in amounts of 2 µg for 48 h using Lipofectamine 3000 (Thermo Fisher Scientific), according to the manufacturer’s instructions. After that, the cells were fixed by incubating them in 4% paraformaldehyde for 15 min at room temperature. After washing the cells with PBS 0.1% Tween-20 (PBST), the cells were treated with PBS 0.5% Tween-20 (PBST) for 10 min. Next, the cells were blocked in 1% skim milk for 20 min at room temperature. The cells were washed with PBS 0.1% Tween-20 (PBST) and incubated overnight at 4 °C with mouse monoclonal FLAG-antibody. Then, the cells were washed with PBST and incubated with Alexa546-conjugated anti-mouse IgG antibody (Thermo Fisher Scientific, Waltham, MA, USA) for 1 h. After the cells had been washed 3 times with PBST, the cell nuclei were counterstained with 1 μg/mL of 4′,6-diamidino-2-phenylindole (DAPI, Thermo Fisher Scientific, Waltham, MA, USA) for 10 min. The cells were finally washed with PBS and mounted on slides with ProLong™ Gold Antifade Mountant (Thermo Fisher Scientific, Waltham, MA, USA). The cells were observed under a confocal microscope (A1R, Nikon, Tokyo, Japan) equipped with 10x, 20x, 40x and 60x lenses; Nomars- 5 ki’s DIC contrast; Hoffman’s modulation contrast; 405-, 488-, 561- and 640-nm lasers; a hybrid scanner; and a resonance scanner (Nikon). The workstation was equipped with Nikon’s Confocal NIS-Elements package. The confocal images were analyzed using the IMARIS 6.0.1 software (Bitplane AG, Oxford, UK).

### 2.6. Electrophoretic Mobility-Shift Assay (EMSA)

Oligonucleotides synthesized by Sigma-Aldrich were used for DNA-binding assays. Sequence information is provided in Appendix A. Generation of double-stranded probes were done by heating equal molar amounts of each of the 5′ to 3′ oligonucleotides with their respective complementary oligonucleotides at 95 °C for 10 min, followed by cooling at room temperature. Next, double-stranded oligonucleotides were labeled with DIG-11-ddUTP using recombinant terminal transferase (20 units/mL) in a final volume of 25 μL, according to the DIG Gel Shift Kit, second generation manufacturer’s instructions (Roche Applied Science, Mannheim, Germany). EMSA was performed according to the manufacturer’s directions. In brief, DNA-binding reactions were set up using 10 μg of a nuclear extract of either WT or mutant proteins. These proteins were mixed with the above-mentioned DIG-labeled oligonucleotides in a DNA-binding buffer containing 1 μg of poly(dI-dC) and 0.1 μg of poly-l-lysine, in a final reaction volume of 20 μL.

### 2.7. Molecular Dynamics (MD) Simulations

The structure of the POUh domain (residues 201 to 278, chain B) of HNF1A was derived from the Protein Data Bank; the ID is 1IC8 [34]. Protein and water molecules within 5 Å of the POUh domain were retained and considered in the initial structure. The N-terminal residue of the protein was capped with an acetyl group to reduce the truncated effect of the POUs domain. Hydrogen atoms were added to the protein and water molecules with the pdb2gmx module of GROMACS [35] under the assumption of the standard protonated state. The simulation system was solvated with TIP3P water molecules [36] and neutralized in a dodecahedron box with a minimum distance of 12.0 Å between the protein and the box edges and with 0.15 M concentrations of Na^+^ and Cl^−^ ions. The AMBER ff14SB parameter set [37] and the parameter set previously reported [38] were employed for the force fields of the protein and Na^+^ and Cl^−^ ions, respectively. The total number of atoms in the box was 29,144. The V259F mutant’s structure was modeled using MODDELER [39]. In the process, residues within 8 Å of the C_β_ atom of V259 could move to avoid any atomic overlap. The mutant system was also prepared as the WT system above. The total number of atoms in the box was 29,154.

All the MD simulations were performed with GROMACS. Ten independent runs were performed as follows: the simulation systems were first subjected to energy minimization with the steepest descent method, followed by the conjugate gradient method. Then, for equilibrating the systems, MD simulations were carried out for 100 ps at 300 K with NVT condition and for another 10 ns at 300 K with NPT condition using Berendsen’s method [40]. Finally, for each system, product runs were carried out for 200 ns at 300 K under NPT condition using the Parrinello–Rahman method [41]. The temperature was maintained with Langevin bath (the time constant for coupling was 2 ps) [42], and the electrostatic interactions were calculated with the particle mesh Ewald method [43]. Non-bonded interactions were cut off at 10 Å, and the bond length including hydrogen atoms was constrained by LINCS method [44] for protein, and the SETTLE method [45] for the water molecules. The integral time step was set to be 2 fs. For analysis, the last 100 ns trajectories were used. The total MD trajectory for analysis was 1 μs.

Residue-wise intra-contact was counted if any of the heavy atoms from a pair of residues was less than 4.5 Å. Then, the differences in contacts between the WT and V259F mutant were calculated by subtracting the contacts of V259F from those of the WT. The last 100 ns of all the 10 trajectories were used for the contact-map calculation. The solvent-accessible surface area was calculated with VMD [46]. The molecular figures were also created with VMD [46].

### 2.8. KD by HNF4A siRNA

For KD, a total 3 × 10^5^ Huh7 cells were plated in 6-well plates and transfected with a 20 nM concentration of either control or HNF4A siRNAs using the Lipofectamine RNAiMAX Transfection Reagent (Thermo Fisher) according to the manufacturer’s instructions and cultured for 48 h in DMEM + 10%FBS medium without antibiotics. The sequences of the siRNAs and primers are listed in Appendix A, and MISSION siRNA Universal Negative Control (SIC-001-s) was obtained from Sigma Genosys (Sigma Genosys Holdings LLC, TX, USA). After 48 h of transfection, the cells were lysed with T-PER for the extraction of whole cell protein, and Western blotting was performed as described above.

### 2.9. RNA-Sequencing (RNA-Seq) and Functional Analysis

Total RNA was extracted from siRNA KD Huh7 cells with the NucleoSpin® RNA kit (MACHEREY-NAGEL, Düren, Germany). For the reverse transcription, 0.5 μg of total RNA was used and the reactions were performed according to the manufacturer’s protocol (EURx, Gdansk, Poland). PCR was performed with the AmpliTaq Gold 360 Master Mix (Applied Biosystems, Waltham, MA, USA) using the GeneAmp PCR System 9700 (Applied Biosystems, Waltham, MA, USA). GAPDH expression was utilized for normalization. RNA-Seq was then carried out via a commercially available service (service ID# F21FTSEUHT1601, BGI, Huada Gene, Wuhan, China). We analyzed the RNA-Seq data for two HNF4A siRNA KD (2 replicates) Huh7 cell samples. The KEGG enrichment pathway and GO bioinformatic analyses were conducted using BGI’s Dr. TOM approach, an in-house customized data-mining system of the BGI. The average of 2 controls and average for the KD (2 siRNA1 and 2 siRNA2) were used to calculate the differential gene expression. The upregulated or downregulated expression of genes was expressed as log2FC, which represents the log-transformed fold change (log2FC = log2[B] − log2[A]).

### 2.10. Statistical Analyses

The data are presented as the means ± standard errors of the means (SEMs) for each group in the experiment. The statistical analyses were performed using a one-way analysis of variance (ANOVA) followed by Tukey’s post hoc tests. P values less than 0.05 were considered to indicate statistical significance. The GraphPad PRISM software version 6 (GraphPad Software Inc., La Jolla, CA, USA) was used for the statistical analysis.

## 3. Results

### 3.1. Somatic Mutations Found in the Functional Domain of HNF1A

Next-generation sequencing has helped to decipher the low-frequency somatic mutations of HCC and identified HNF1A as a candidate driver gene [29]. HNF1A mutations mostly located in the POU domain of HNF1A, identified in the ICGC database, are presented here (https://dcc.icgc.org/, accessed on 12 October 2020). The POU-domain mutations reported in different liver cancer projects are listed in Figure 1A and Table 1. The data suggest that HNF1A mutations in these regions may have an impact on hepatocarcinogenesis. In our study, we performed functional analyses of three mutations (Y122C, R229Q and V259F) located in the POUs and POUh domains (Figure 1A). From an evolutionary perspective, the mutant amino acid residues are strictly conserved among various species (Figure 1B); the asterisks in red specify the locations of the POU domain mutations (Y122C, R229Q and V259F). The conserved domains among the different species (humans, mice, bovines and zebrafish) are highlighted in red, and the domains that we functionally analyzed are 100% conserved throughout the different species. The mutations in such evolutionarily conserved elements might have a strong effect on the protein function and warrant further investigation. 

### 3.2. HNF1A Mutants Display Reduced Transcriptional Activity and Decreased Binding Ability

It has been reported that HNF1A mutations affect DNA binding and reduce the transcriptional activity. However, there are few reports on the functional analysis of disease-associated mutations in HNF1A [30,32]. Thus, we sought to determine how the novel mutations found in the POU domain affected the properties of the mutant proteins and impaired the transcriptional ability of HNF1A. To evaluate the effects of these somatic mouse (Y122C, R229Q and V259F) and human HNF1A (Y122C and V259F) mutations in the POU domain, we examined the transcriptional activity of those mutants found in liver cancer patients. We compared the ability of the human and mouse HNF1A mutant proteins to transactivate HNF1A-responsive elements containing the *HNF4A* P-1 promoter (Figure 2A,B).

The overexpression of human and mouse WT HNF1A stimulated the transcription of HNF1A-responsive element-containing promoters; however, Y122C, R229Q human and Y122C mouse mutations resulted in a decreased transactivation function for HNF1A toward *HNF4A* P1 (Figure 2A,B). More importantly, the mouse and human HNF1A V259F mutations completely lost their transcriptional activity in all cases (Figure 2A,B). In our study, we found a similar effect of the HNF1A mutants on the *HNF4A* P2 promoter (Appendix A). With Huh7 cells, which endogenously express HNF1A, we found that HNF1A WT had higher transcriptional activity, but both mutations (Y122C and V259F) resulted in reduced transcriptional activity for the *HNF4A* P1 promoter (Figure 2C), and similar activity was also found in the case of the *HNF4A*-P2 promoter (Appendix A). These results are consistent with a previous study indicating that MODY3-associated mutants displayed reduced transcriptional activity for their target promoter [47,48]. Therefore, our functional analysis revealed that the mutations in the POU domain cause reduced HNF1A transcriptional activity, suggesting that the mutations located in this domain merit further study.

As most of the somatic mutations analyzed in our study are localized in the POU domain (Figure 1A), we investigated the DNA-binding ability of the mutant HNF1A proteins. Furthermore, reduced transcriptional activity suggests that mutations may directly affect the DNA-binding ability of HNF1A. Using the EMSA, we measured the DNA-binding affinity of WT and mutant HNF1A proteins. We found a clear correlation between the effects of these mutations on HNF1A transcriptional activation and DNA binding. HNF1A Y122C, R229Q and V259F mutants exhibited markedly reduced binding to the *HNF4A* promoter compared to the WT HNF1A (Figure 2E), whereas the WT and mutant HNF1A proteins were expressed equally, as demonstrated by Western blot (WB) analysis (Figure 2F). Changes in the nuclear localization of proteins may affect transcriptional activity. Therefore, we analyzed whether mutations of HNF1A (Y122C, R229Q and V259F) affected its proper nuclear localization ability. IFC staining revealed that both the WT and mutant HNF1A were localized in the nuclei of HEK293 cells (Figure 2D). Thus, our findings strongly suggest that HNF1A Y122C, R229Q and V259F mutants have reduced transcriptional activity due to the loss of their ability to bind to *HNF4A* promoter regions, and these are related to the loss of HNF4A expression and function. Notably, the RNA-Seq data obtained from The Cancer Genome Atlas (TCGA) database of cancer patients showed that the expression of *HNF4A* and *HNF1A* mRNA is significantly correlated in many cancer types (Appendix A). These results suggest that HNF1A and HNF4A are involved in a cross-regulatory network, and if a loss-of-function mutation occurs in one, it may lead to the reduced expression of the other. In our previous study, we found that HNF4A Zn-finger mutations resulted in a similar phenotype and that the *HNF1A* promoter could not bind with the HNF4A G79C mutant, partially due to the disrupted fluctuation of the protein structure ([26], Appendix A). Therefore, we further investigated whether this type of structural change occurred when the HNF1A POU domain was mutated.

### 3.3. Dynamics of the HNF1A V259F Mutant Revealed That the Mutation Affects Protein Stability and Causes Rearrangement in the N-Terminal Region

V259 is located in the POUh domain, related to the DNA-binding region of HNF1A, and is thus considered functionally important (Figure 3A). It should be noted that V259 is not directly involved in protein–DNA interactions, but the mutation has been found to reduce the binding affinity. To examine the impact of the mutation, we conducted 10 independent all-atom MD simulations with explicit solvent models for each of the WT and V259F proteins. The root-mean-square fluctuations (RMSFs) showed that the fluctuations of the structures were similar, except for the N-terminal region, in which they differed (Figure 3B). V259F had a significantly larger fluctuation than the WT at the N-terminal but not in other regions, including the DNA-recognition helix (residues 260 to 274) and mutation site.

We further investigated why this large fluctuation occurred in the N-terminal region. The residue-wise contact map illustrates the changes in the interaction between the two residues. The map shows that V259F lost several key interactions: the hydrophobic interaction of V259–V264 and electrostatic and/or hydrophobic interactions of N237–L258, K205–S256, N237–N257 and R203–S256 (Figure 3C,D). The loss of these interactions destabilized the hydrophobic packing formed around V259 in the WT.

As seen in Figure 3E,F, in the WT structure, the 259th residue Val was nearly always shielded from the solvent. In the mutant structure, the mutated Phe was often exposed to the solvent. We observed a correlation between the solvent-accessible surface area of the Phe and the fluctuation in the N-terminal region (Figure 3E,F). This suggests that the N-terminal region managed to shield the Phe from the solvent, but that conformation was unstable, thereby causing the large fluctuation in the N-terminal region. These large conformational changes in the N-terminal region result in the loss of DNA interactions by R203 and K205, reducing the DNA-binding affinity. Furthermore, this fluctuation affects the arrangement of the POUh and POUs domains, both of which bind to DNA.

### 3.4. siRNA KD of HNF4A Causes Differential Gene Expression and Overrepresented Pathways

Overall, the results suggest that POU domain mutations of HNF1A downregulate *HNF4A* gene expression. Therefore, to mimic the HNF1A mutation phenotype in transcription networks, we performed siRNA-mediated KD of HNF4A. Two pairs of oligonucleotides encoding HNF4A-specific siRNAs were designed to silence HNF4A expression. After 48 h of transfection, the HNF4A levels were significantly decreased in Huh7 cells through HNF4A siRNA treatment (Figure 4A). We also examined the changes in HNF4A protein levels in Huh7 cells, which endogenously express high levels of the HNF4A protein. The HNF4A siRNA markedly reduced the HNF4A protein levels as compared with the controls (Huh7 cells transfected with the control siRNA; Figure 4B). HNF4A is a known tumor suppressor, regulating the transcription of a myriad of genes [10,25,26]. To further understand the effect of KD on the mechanism underlying HNF4A’s tumorigenic function, RNA-Seq analysis was performed to evaluate the genome-wide gene expression profile in HCC cells after HNF4A KD. RNA-Seq data analysis revealed that 748 genes were differentially expressed in the HNF4A KD cells (Figure 4C). We found a distinct difference in the global gene expression profile in control versus KD cells; among 748 genes, 311 genes were downregulated and 437 were upregulated (Figure 4D). The KD of HNF4A resulted in the down- and upregulation of many genes known to be involved in transcriptional regulation (Appendix A). Kyoto Encyclopedia of Genes and Genomes (KEGG) pathway enrichment analysis showed that the most overrepresented pathways were the Hippo signaling pathway, and the lipid and cholesterol metabolic pathways (Figure 4E). Gene ontology (GO) analyses revealed that the genes were largely involved in biological processes, such as lipid and cholesterol metabolism, and extracellular matrix organization (Figure 4F). GO analysis also showed that the genes were involved in molecular functions, such as binding activity (e.g., protein, cholesterol, actin filament, and signaling receptor binding; Appendix A). The protein–protein interaction analysis of the downregulated genes revealed that HNF4A downregulation also caused the downregulation of its target proteins, which are involved in lipid and cholesterol metabolism (Figure 4G).

## 4. Discussion

Recent advances in NGS technologies have identified major cancer-driving genes in the liver, and their mutations are related to liver carcinogenesis [5,49]. Previously, HNF1A mutations were identified in diabetes, and their functional effect was validated [31,50,51]; however, only very few studies have suggested that HNF1A mutations identified in HCC are associated with the development and progression of HCC [32]. Interestingly, we found that HNF1A was one of the genes commonly found to be mutated in HCC according to the ICGC database (https://dcc.icgc.org/, accessed on 12 October 2020), and many mutations are accumulated in the POU domain of HNF1A (Table 1). On the other hand, HNF4A is also known as a major tumor suppressor, and its expression is tightly regulated by HNF1A [26,27,28]. These findings suggested that the role of HNF1A POU domain mutations in the regulation of HNF4A expression in the context of HCC molecular pathology merited investigation. In this study, we demonstrated the functional effect of POU domain mutations of HNF1A on HNF4A gene regulation and investigated their effects on alterations in transcriptional networks through the dysregulation of HNF4A gene expression. 

The HNF family harbors common features such as DNA-binding and transactivation capabilities that account for its functional diversity [29,52]. HNF family gene mutations are mostly known to occur in the functional domain of the protein and inhibit the protein’s activity by affecting its DNA-binding affinity and protein conformation [26,53]. Our study presents a systematic analysis of the ICGC database of HNF1A transcription factor (DNA-binding protein) mutations in the POU domain (Table 1). Notably, the HNF1A mutations reported in this domain are highly conserved among different species (Figure 1B). We studied three substitution mutations (Y122C, R229Q and V259F) in the POUs and POUh domains (Figure 1A). The results imply that the HNF1A mutations we identified in the POU domain are pathogenic mutations that strongly affect protein function and augment the risk of the initiation of liver cancer development. Previous studies have demonstrated that the R271W and S247T mutations of HNF1A located in the POUh domain impair HNF1A’s transcriptional activity to transactivate the *HNF4A* promoter [48,54]. These results are consistent with the data from our study, in which we found impaired transcriptional activity of HNF1A Y122C, R229Q and V259F mutants in the regulation of *HNF4A* promoter activity. Moreover, the HNF1A Q511L mutation was reported to reduce the function of HNF1A to regulate *HNF4A* promoter activity as well as to inhibit the proliferation, migration, and invasion of HCC cells [32]. Therefore, our results suggest that reduced *HNF4A* promoter activity caused by HNF1A POU-domain mutations may play a role in HCC development. Loss-of-function mutations caused by substitution or deletion represent the majority of functionally characterized MODY mutations [31,34]. In fact, several functionally validated HNF1A mutations have been found in MODY patients [47,48,55]. Apart from the mutations verified in this study, we found several HNF1A mutations located in the POU domain (Table 1). While our study emphasizes the importance of POU-domain mutations of HNF1A, further functional studies are needed to verify the mutations found in different countries. Similarly, since HNF1A mutations are commonly found in MODY patients, it is clinically important to verify the risk of liver cancer development in MODY patients.

It is known that in mice the hepatocyte-specific deletion of HNF1A leads to the spontaneous development of HCC due to fatty liver without cirrhosis [11]. Moreover, the hepatocyte-specific deletion of HNF1A in mice leads to non-alcoholic steatohepatitis (NASH) and HCC [11]. Similarly, the KO/KD of the major HNF1A target gene *HNF4A* is known to play a role in liver oncogenesis or HCC [56,57,58], suggesting that both HNF1A and HNF4A are responsible for maintaining liver homeostasis, and the disruption of their function may lead to liver pathologies and HCC. In our study, we observed that HNF1A Y122C, R229Q and V259F mutations significantly decreased the transcriptional activity regarding the regulation of the *HNF4A* gene and reduced the DNA-binding capacity of HNF1A for the *HNF4A* promoter. Conversely, the HNF4A G79C mutation reduced the ability of HNF4A to bind to the *HNF1A* promoter (Appendix A). We and others have suggested that HNF1A and HNF4A are involved in a regulatory network [26,59,60] and that their gene expression is tightly correlated [60,61] (Appendix A); as such, pathogenic mutations in either the *HNF1A* or *HNF4A* gene may increase the risk of HCC by reducing their expression. In fact, a few studies have revealed that the HNF1A–HNF4A axis is an important pathway for the control of liver homeostasis and that its disruption can cause liver cancer. However, further in vivo studies are needed to clarify the importance of these possible pathogenic mutations in HCC. 

The loss-of-function V259F mutation in HNF1A was subjected to rigorous structural and stability analyses to identify its deleterious effect. MD simulations allowed us to elucidate the dynamic nature of the protein–DNA interaction when the mutation occurred at an atomic level (Figure 3D). As demonstrated by Sneha et al. [53], a higher RMSF is associated with reduced stability, consistent with our observation that the V259F mutant complex exhibited a greater fluctuation pattern (Figure 3C), which was correlated with a reduction in the number of intermolecular hydrogen bonds formed in the V259F mutant complex compared with the WT HNF1A complex (Figure 3A,C). It is known that proteins have arginine residues on their surfaces, which greatly increases the proteins’ stability [62]. By contrast, the rearrangement of arginine and lysine residues results in reduced stability and negatively affects the protein function. However, according to our WB experiment, the HNF1A protein and mutants are equally expressed. The complete loss of DNA binding for V259F suggests that valine is an essential base that is important for DNA interaction and DNA-binding affinity. It has been reported that the disruption of helix 3 (residues 260 to 274) during substitution mutations could cause a conformational change in the protein and affect the protein’s function [47]. In line with this, it is postulated that V259F changes the conformation of the HNF1A protein’s structure and gives rise to an unstable structure in the N-terminal region. Altogether, we conclude that the large conformational changes in the N-terminal region, but not the change in protein stability, resulted in the loss of DNA interactions by R203 and K205, reducing the DNA-binding affinity.

In this study, we found that loss-of-function mutations of the HNF1A POU domain trigger a reduction in HNF4A gene expression. However, the molecular mechanism through which the loss of function may cause disrupted gene expression and, therefore, promote HCC at the molecular level remains to be understood. To determine the molecular mechanisms, we performed a global gene expression analysis in the condition of HNF4A KD. The top seven downregulated genes we found were HPR, PKLR, PLAU, SOAT2, IYD, OTC and ASGR1. Notably, two metabolic genes, OTC and ASGR1, were previously identified as potential prognostic biomarkers in HCC [63,64]. Several studies have suggested that OTC deficiency in the liver leads to the build-up of ammonia, which causes chronic liver damage, and this is a major risk factor of HCC [65]. Moreover, increased liver fibrosis has been observed in heterologous OTC-KO mice [66]. Additionally, OTC overexpression has been shown to inhibit HCC cell proliferation [63]. Therefore, low OTC expression may enable tumor cells to increase ammonia accumulation, representing a loss of function of the tumor-specific metabolism of OTC. Gu et al. [64] reported that ASGR1 overexpression reduced hepatoma cell migration and invasion by interacting with LASS2. Here, we found that the expression of a serum glycoprotein homeostasis regulator, ASGR1, was downregulated in HNF4A KD cells, suggesting that HNF4A positively regulates ASGR1 expression in HCC cells. Therefore, our result is consistent with the previous report and suggests the role of ASGR1 as a tumor suppressor in HCC [67]. Furthermore, it has been reported that IYD overexpression suppressed Huh7 cell growth by inhibiting glycolysis in HCC cells [68]. Therefore, the downregulation of IYD in HNF4A KD cells is considered as a key driver in HCC malignancy, especially when both HNF1A and HNF4A have loss-of-function activity. However, the contribution of IYD in relation to HNF-family genes to tumorigenesis in the liver has not been investigated yet, and further studies are needed. HPR, PKLR and PLAU have been reported to be overexpressed in breast cancers and esophageal squamous cell carcinoma [69,70,71]. Conversely, our study showed the downregulation of those genes. It is possible that these genes might be tissue specific, and their downregulation may promote the transition from liver damage to hepatocarcinogenesis and enhance HCC progression in the presence of the loss of function of HNF4A/1A in HCC, but further studies are needed to validate this hypothesis. On the other hand, several genes downregulated in the HNF4A gene network are involved in lipid and cholesterol metabolism, and the downregulation of these genes may promote cancer development. GATA4, APOC3, APOA1 and FOXO1 were found to be downregulated in Huh7 HNF4A KD cells, which were previously reported as cholesterol and lipid metabolism related genes [72,73,74]. Hepatocyte-specific Gata4-KO mice developed enlarged livers with a proliferative precursor phenotype [75], thus play a role in liver cancer development. HNF4A KD in Huh7 cells reduced SOAT2 mRNA expression. It was previously reported to reduce lipogenesis and de novo cholesterol synthesis in HNF4A KD mice through the inhibition of SOAT2 expression [76]. Based on the overall findings, it is suggested that HNF4A is one of the master regulators of lipid and cholesterol homeostasis, and the disruption of the function of HNF1A caused by mutations may trigger liver cancer development and progression due to the disruption of lipid and cholesterol homeostasis as well as key liver functions such as ammonia and glycoprotein homeostasis. Further in vitro and in vivo studies are required to assess the mutational effect of HNF1A on HCC development.

In conclusion, our study provides new insights into the tumorigenic mechanisms related to HNF1A mutations in the liver. In HCC, our tested mutations in the POU domain of HNF1A that resulted in a loss of function regarding activity in the regulation of the *HNF4A* promoter caused a reduction in HNF4A mRNA expression, with the disruption of lipid metabolism, through the dysregulation of transcriptional networks. Additionally, our findings suggest that HNF1/4A is one of the master regulators of liver cell differentiation and lipid homeostasis and support the idea that any disruption of this transcriptional network may cause liver cancer development and progression.

## Figures and Tables

**Figure 1 genes-13-00413-f001:**
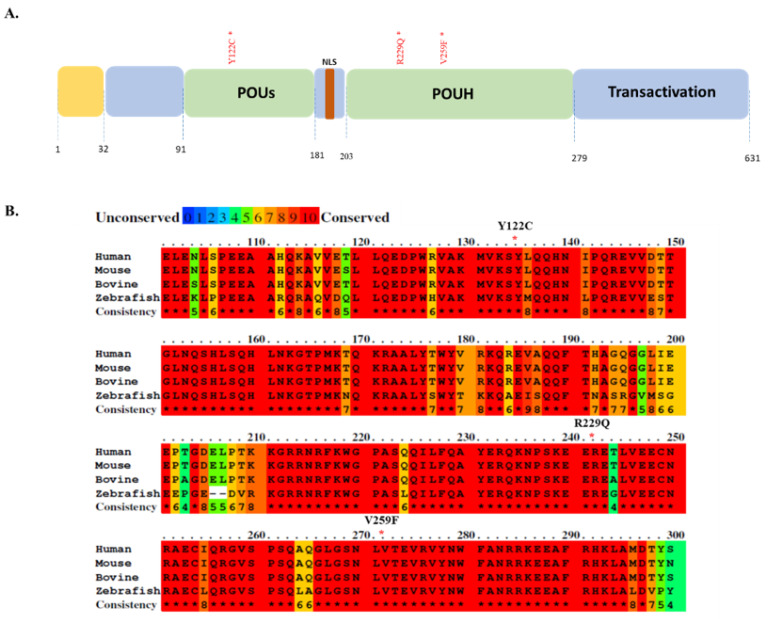
(**A**) Positions of novel mutations are indicated in the human HNF1A protein structure (Pit1, Oct1 and Unc1 (POU) domain-Green; POU homeodomain (POUh) and POU-specific (POUs)). (**B**) Alignment of the human, mouse, bovine and zebrafish HNF1A amino acid sequences and mutations found in the POU domain. Red color denotes highly conserved (100%) elements among the species. The mutations (indicated by red asterisks) in the POU domain of HNF1A are highly conserved in species.

**Figure 2 genes-13-00413-f002:**
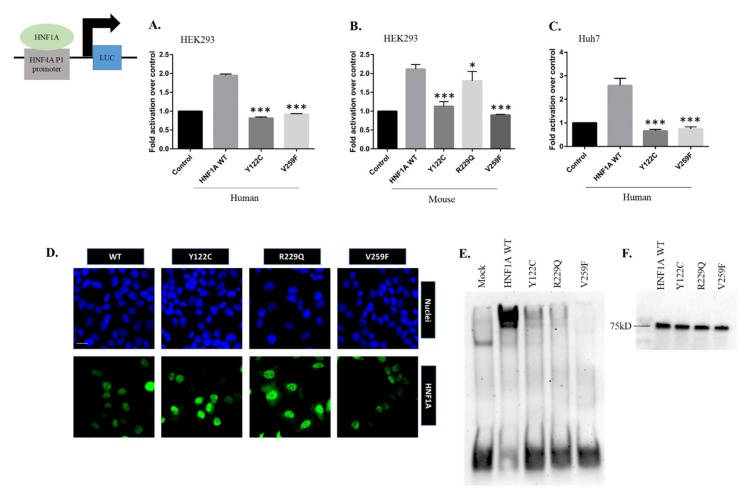
The ability of the (**A**) human WT and mutant HNF1A and (**B**) mouse WT and mutant HNF1A to transactivate *HNF4A* P1 when overexpressed in HEK293 cells. (**C**) The ability of the human WT and mutant HNF1A to transactivate the target promoter (*HNF4A* P1) when overexpressed in Huh7 cells. The cells were co-transfected with the indicated luciferase reporters and either an empty expression vector (serving as a control) or expression vectors (100 ng) for the indicated HNF1A vectors in 24-well culture plates. The bars indicate the fold activation of HNF1A WT and mutants (vs. control) on target promoters. The corresponding promoter activity is reported as fold activation over control (±SEM, *n* = 3–4). The data reported represent the averages of three experiments, each conducted in duplicate. (*, *p* < 0.05; ***, *p* < 0.001). (**D**) Cellular localization of WT and mutant HNF1A was visualized in HEK293 cells using IFC staining. The nuclei were stained with DAPI, and the images were taken at 20× magnification. (**E**) Electrophoretic mobility shift assay (EMSA) was used to assess the binding of WT or mutated HNF1A nuclear proteins to a double-stranded oligonucleotide corresponding to the consensus HNF1A-binding elements of the *HNF4A* promoter region. The HNF1A V259F mutant displayed markedly reduced binding to the *HNF4A* promoter region for all the experiments. (**F**) HEK293 cells were transfected with expression vectors encoding HNF1A WT or the indicated mutants. WB analysis showed that all proteins were similarly expressed.

**Figure 3 genes-13-00413-f003:**
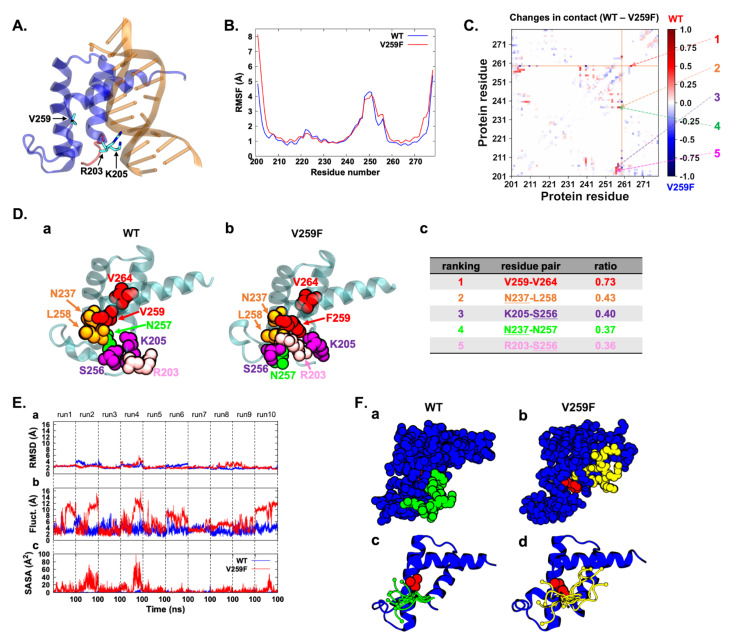
(**A**) X-ray crystallographic structure of the POUh domain (blue) binding with DNA (orange; PDB ID: 1IC8). The backbone of the N-terminal region is highlighted in red. The residues V259, R203 and R205 are depicted by the stick model. (**B**) Fluctuations of the WT (blue) and V259F mutant (red) proteins. The RMSF of the backbone was plotted. The fluctuations were calculated using the last 100 ns of 10 runs (1 μs in total). (**C**) Residue-wise contact-map difference between the WT and V259F mutant proteins. The contacts were calculated using the last 100 ns of 10 runs (1 μs in total). The colors denote the ratios of contact in the simulation time. The five most contacted residue pairs in WT proteins are denoted with numbers. We defined contact as any heavy atom of the residue-pair within 4 Å. (**D**) (a) The WT structure closest to the averaged structures over 1 μs trajectories. The protein backbone is drawn in a ribbon representation (cyan). The heavy atoms of the five most contacted residue pairs are depicted with a space-filling model; see also (c). (b) The V259Fmutant structure closest to the averaged structures over 1 μs trajectories. The heavy atoms of the five most contacted residue pairs in the WT are depicted in the space-filling model. The residue pairs in contact in the WT were completely lost. (c) Contact ratio of residue pairs in the 1 μs trajectories. The residues in the list are shown in (a,b) with distinct colors. Underlined residues appeared twice in the list. (**E**) (a) Backbone-RMSD of the whole POUh domain except for the N-terminal region (residues 201 to 206) against the X-ray crystallographic structure. (b) Fluctuation of the N-terminal region. The fluctuation was calculated using the RMSD-fitted structures of (a). (c) Solvent-accessible surface area (SASA) of the sidechain atoms of residue at position 259: Val of the WT is depicted in blue, and mutated residue Phe, in red. Dotted lines show the boundaries of runs. (**F**) A typical snapshot of the WT (a) and V259F mutant proteins (b) in the trajectory. The 259th residue and the N-terminal region are depicted by red and green/yellow, respectively. (c) The conformations in the N-terminal region of the WT. The images depict 10 structures taken from the last snapshots of 10 runs in green, and V259 is denoted by red in the space-filling model. (d) The conformations in the N-terminal region of the V259F mutant. The images depict 10 structures taken from the last snapshots of 10 runs in yellow, and F259 is denoted by red in the space-filling model.

**Figure 4 genes-13-00413-f004:**
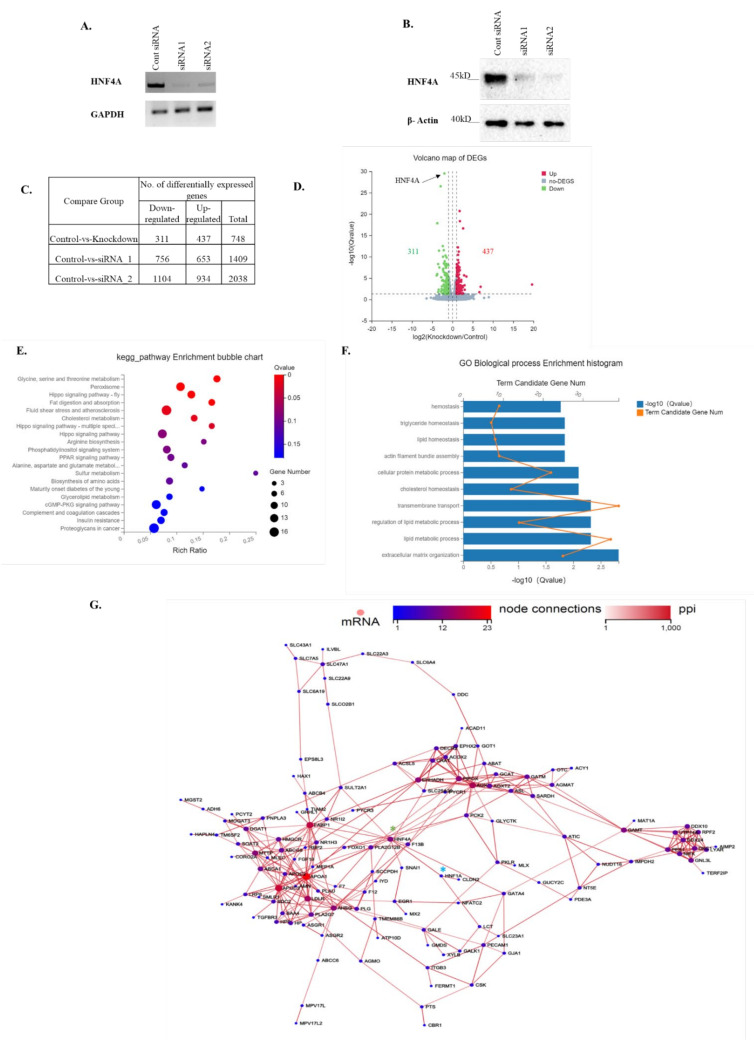
HuH7 cells were transfected with 20 nM concentrations of either control siRNA (SIC; MISSION® siRNA Universal Negative Control #1) or HNF4A-specific siRNAs. (**A**) The KD efficiency was verified by RT-PCR. (**B**) Western blot analysis shows that HNF4A expression was decreased in HNF4A-siRNA-treated HuH7 cells. (**C**) The number of differentially expressed genes (DEGs) in HNF4A-knockdown (KD) cells. (**D**) Volcano plot map representing DEGs in KD cells. Red dots represent upregulated genes, green dots show downregulated genes, and gray represents non-DEGs in KD cells. (**E**) Top 20 KEGG pathways in KD cells. (**F**) GO analyses of the top 10 biological processes. (**G**) Protein–protein interaction of downregulated genes. Green asterisk indicates HNF4A, and blue asterisk indicates HNF1A in the network.

**Table 1 genes-13-00413-t001:** HNF1A mutations located in POU domain found in liver cancer patients.

Mutation ID	Genomic DNA Change	Type	Consequences	Project in Which Mutation Observed	Conservation among Species
MU854410	chr12:g.121432028G>T	Single base substitution	V259F	LINC-JP	YES
MU837628	chr12:g.121426674A>G	Single base substitution	Y122C	LINC-JP	YES
MU81565444	chr12:g.121431445G>T	Single base substitution	A217S	LICA-CN	YES
MU85877851	chr12:g.121426663G>T	Single base substitution	M118I	LICA-CN	YES
MU20638	chr12:g.121431482G>A	Single base substitution	R229Q	LICA-FR	YES
MU29769426	chr12:g.121431410A>C	Single base substitution	K205T	LICA-FR	YES
MU82396333	chr12:g.121426664G>T	Single base substitution	V119F	LICA-CN	YES
MU602436	chr12:g.121426701G>T	Single base substitution	R131L	LICA-CN	YES
MU29793014	chr12:g.121431983A>G	Single base substitution	R244G	LIHC-US	YES
MU29769474	chr12:g.121431977A>T	Single base substitution	I242F	LICA-FR	Not in Zebrafish
MU128970370	chr12:g.121426782A>G	Single base substitution	K158R	LIHC-US	YES
MU85320917	chr12:g.121431501G>T	Single base substitution	E235D	LICA-CN	YES
MU128971993	chr12:g.121431424T>A	Single base substitution	S210T	LIHC-US	YES
MU29496420	chr12:g.121432040C>G	Single base substitution	R263G	LIHC-US	YES
MU29433874	chr12:g.121432014T>A	Single base substitution	L254Q	LIHC-US	YES
MU29746856	chr12:g.121431506A>G	Single base substitution	N237S	LIHC-US	YES
MU822656	chr12:g.121431466C>G	Single base substitution	P224A	LIAD-FR	YES
MU822434	chr12:g.121431413G>T	Single base substitution	W206L	LIAD-FR	YES
MU822864	chr12:g.121432067C>A	Single base substitution	R272S	LIAD-FR	YES
MU823044	chr12:g.121432041G>T	Single base substitution	R263L	LIAD-FR	YES

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
