# Peer review of "HNF1A POU Domain Mutations Found in Japanese Liver Cancer Patients Cause Downregulation of HNF4A Promoter Activity with Possible Disruption in Transcription Networks"

_genes, 2022, doi:10.3390/genes13030413_

Round 1
Reviewer 1 Report
The manuscript is of high quality regarding problem described, study design, used sophisticated methodology, presentation the results and discussion. I have no concerns.
presentation of results and description. It preseants current knowledge on th problem, discusion is clear.
The manuscript is prepeared with clearity and care. Its result can be taken into considerations by clinicians.
Provide origin, passage number, and information on routine tests you use to check morphological and biochemical cell line properties
Author Response
>The manuscript is of high quality regarding problem described, study design, used sophisticated methodology, presentation the results and discussion. I have no concerns. presentation of results and description. It preseants current knowledge on th problem, discusion is clear. The manuscript is prepeared with clearity and care. Its result can be taken into considerations by clinicians.Provide origin, passage number, and information on routine tests you use to check morphological and biochemical cell line properties
Thank you for your high evaluation on our manuscript. We also hope our manuscript can be read by clinicians. HEK293 cells were purchased from ATCC. HuH7 cells were donated from the lab of Dr. Hidewaki Nakagawa, RIKEN IMS. The information was described in the material and method section. These cells were used for the experiments before reaching their passage 30. We culture only these two human cell lines at the lab, and we can recognize the cells by checking their morphology (in case of HuH7 cells lipid droplet is accumulated in the cells) and HNF4A protein expression using Western blotting (HNF4A is expressed only in HuH7 cells). As you pointed out, we agree with the good recognition of the cells using proper markers and we verify the cells when we perform the experiments. Thank you very much for your constructive comments.
Reviewer 2 Report
This is an interesting paper presenting a data regarding a few genes to be associated with hepatocarcinogenesis. However, this paper misses a one crucial point – what was the rationale for cherry-picking those genes. In addition, authors have used some exaggerating words when referred to their findings. Although your findings are interesting there could be difficulties in receiving similar outcomes by following your experimental plan in other labs or clinics. Finally, my major point – please provide strong rationale for choosing those genes and amend discussion accordingly. Provide more comparisons in your discussion to other studies.
All acronyms should be spelled first, otherwise please provide a list with all abbreviations used.
The introduction chapter is disorganised. Please more correctly elaborate and add references to each statement. Some parts, e.g regarding third most common cancer-related death, or regarding risk factors are incorrect. So please amend.
Please provide rs code(s) for V259F mutation and other listed throughout the manuscript
Author Response
This is an interesting paper presenting a data regarding a few genes to be associated with hepatocarcinogenesis.
>However, this paper misses a one crucial point – what was the rationale for cherry-picking those genes.
>Finally, my major point – please provide strong rationale for choosing those genes and amend discussion accordingly. Provide more comparisons in your discussion to other studies.
Thank you for your helpful comments to our manuscript to improve its quality. To explain the rational of studying the effect mutation of HNF1A and its downstream target gene HNF4A, we newly added the sentences in the Introduction and Discussion as follows. Additionally, we discussed about consistency and inconsistency about our study and already existing studies in the Discussion as follows.
Introduction)
-Why we focused on HNF1A mutations in HCC?
Besides these major driver gene mutations, other driver and tumor suppressor genes with well-established roles in liver function have been found. Among many gene mutations found through NGS-based mutational detection, hepatocyte nuclear factor 1A (HNF1A) was found to be a frequently mutated gene, one of the top 20 mutated genes in hepatocellular carcinoma (HCC) reported in the International Cancer Genome Consortium (ICGC) database (https://dcc.icgc.org/). HNF1A, a liver-enriched transcription factor, is present in embryonic tissues and plays a pivotal role in cellular differentiation and organ development [8]. HNF1A acts synergistically with HNF4A to regulate gene expression in various tissues, including the intestine and kidney [9,10]. In addition to its function in liver development, a recent study demonstrated that HNF1A knockout (KO) mice developed HCC due to fatty liver [11]. Moreover, in the HCC microenvironment, HNF1A inhibits Wingless-related integration site (Wnt) and nuclear factor kappa-B (NF-κB) signaling during metastasis and hepatocarcinogenesis [12,13]. On the other hand, the overexpression of HNF1A suppressed the proliferation of HCC and induced the expression of liver-specific genes in HCC cells, which caused cell cycle arrest [14]. These results support the idea that the dysfunction of HNF1A may cause hepatocarcinogenesis and HCC progression. However, while the role of HNF1A in different cancers has recently been examined [15-17], only a few studies have demonstrated a critical link between HNF1A mutations and the development of liver cancer.
-Why we studied on HNF1A-HNF4A axis in HCC?
Structurally, HNF1A has three domains: a dimerization domain, a DNA-binding domain, and a transactivation domain. The central DNA-binding domain is composed of a Pit-Oct-Unc (POU) homeodomain (POUh) and POU-specific (POUs) domain and is indispensable for efficient transcriptional activity [18]. HNF1A interacts with target DNA as a homodimer or heterodimer with HNF1B to regulate glucose metabolism, lipid metabolism, and detoxification [19-21]. HNF1A occupies the HNF4A promoter region and upregulates its expression as positive feedback [22]. Accordingly, a reduction in HNF4A has been associated with the reduced expression of HNF1A in young mice [23]. Similarly, HNF4A and HNF1A, together, form a network that regulates the expression of each as well as multiple liver-specific genes [22,24,25]. Additionally, our group reported for the first time that HNF4A G79C, F83C, and M125I mutations are loss-of-function mutations found in liver cancer patients, leading to a reduction in HNF1A gene expression and concomitantly, an increased risk of HCC development [26]. Several studies have demonstrated that HNF1A and HNF4A reciprocally regulate each other’s expression through DNA-binding-dependent and independent (protein–protein interaction) mechanisms [27,28]. These findings suggest that both HNF1A and HNF4A are critical regulators of liver function, and their dysfunction leads to liver cancer development. However, unlike for HNF4A mutations, the effects of HNF1A mutations on HNF4A gene regulation and HCC development remain elusive. Notably, much like HNF4A mutations, the ICGC and The Cancer Genome Atlas (TCGA) have reported mutations in the DNA-binding domain of HNF1A [29,30]. Previous studies have established that HNF1A mutations are associated with hepatocellular adenomas and maturity-onset diabetes of the young type 3 (MODY3) [30,31]. P112L and Q466X mutations of HNF1A have been associated with MODY [31]. Although mutations of HNF1A Q511L, E32*, and L214Q have also been identified in HCC [30,32,33], the effect of HNF1A POU domain mutation on the regulation of HNF4A and its downstream molecular mechanism to trigger HCC remain unknown.
Discussion
Recent advances in NGS technologies have identified major cancer-driving genes in the liver, and their mutations are related to liver carcinogenesis [5,49]. Previously, HNF1A mutations were identified in diabetes, and their functional effect was validated [31,50,51]; however, only very few studies have suggested that HNF1A mutations identified in HCC are associated with the development and progression of HCC [32]. Interestingly, we found that HNF1A was one of the genes commonly found to be mutated in HCC according to the ICGC database (https://dcc.icgc.org/ ), and many mutations are accumulated in the POU domain of HNF1A (Table 1). On the other hand, HNF4A is also known as a major tumor suppressor, and its expression is tightly regulated by HNF1A [26-28]. These findings suggested that the role of HNF1A POU domain mutations in the regulation of HNF4A expression in the context of HCC molecular pathology merited investigation. In this study, we demonstrated the functional effect of POU domain mutations of HNF1A on HNF4A gene regulation and investigated their effects on alterations in transcriptional networks through the dysregulation of HNF4A gene expression.
Newly added comparisons between our study and other studies in the Discussion
as follows and highlighted as yellow in the main text:
Comparison1 (in Discussion)
Previous studies have demonstrated that the R271W and S247T mutations of HNF1A located in the POUh domain impair HNF1A’s transcriptional activity to transactivate the HNF4A promoter [48, 54]. These results are consistent with the data from our study, in which we found impaired transcriptional activity of HNF1A Y122C, R229Q and V259F mutants in the regulation of HNF4A promoter activity. Moreover, the HNF1A Q511L mutation was reported to reduce the function of HNF1A to regulate HNF4A promoter activity as well as to inhibit the proliferation, migration, and invasion of HCC cells [32]. Therefore, our results suggest that reduced HNF4A promoter activity caused by HNF1A POU-domain mutations may play a role in HCC development.
Comparison2 (in Discussion)
Moreover, the hepatocyte-specific deletion of HNF1A in mice leads to non-alcoholic steatohepatitis (NASH) and HCC [11]. Similarly, the KO/KD of the major HNF1A target gene HNF4A is known to play a role in liver oncogenesis or HCC [57-59], suggesting that both HNF1A and HNF4A are responsible for maintaining liver homeostasis, and the disruption of their function may lead to liver pathologies and HCC. In our study, we observed that HNF1A Y122C, R229Q and V259F mutations significantly decreased the transcriptional activity regarding the regulation of the HNF4A gene and reduced the DNA-binding capacity of HNF1A for the HNF4A promoter. Conversely, the HNF4A G79C mutation reduced the ability of HNF4A to bind to the HNF1A promoter (Supplementary File 1, Figure S2B). We and others have suggested that HNF1A and HNF4A are involved in a regulatory network [26,60,61] and that their gene expression is tightly correlated [61,62] (Supplementary File 1, Figure S3); as such, pathogenic mutations in either the HNF1A or HNF4A gene may increase the risk of HCC by reducing their expression. In fact, a few studies have revealed that the HNF1A–HNF4A axis is an important pathway for the control of liver homeostasis and that its disruption can cause liver cancer. However, further in vivo studies are needed to clarify the importance of these possible pathogenic mutations in HCC.
Comparison3 (in Discussion)
The complete loss of DNA binding for V259F suggests that valine is an essential base that is important for DNA interaction and DNA-binding affinity. It has been reported that the disruption of helix 3 (residues 260 to 274) during substitution mutations could cause a conformational change in the protein and affect the protein’s function [64]. In line with this, it is postulated that V259F changes the conformation of the HNF1A protein’s structure and gives rise to an unstable structure in the N-terminal region. Altogether, we conclude that the large conformational changes in the N-terminal region, but not the change in protein stability, resulted in the loss of DNA interactions by R203 and K205, reducing the DNA-binding affinity.
Comparison4 (in Discussion)
The top seven downregulated genes we found were HPR, PKLR, PLAU, SOAT2, IYD, OTC and ASGR1. Notably, two metabolic genes, OTC and ASGR1, were previously identified as potential prognostic biomarkers in HCC [65, 66]. Several studies have suggested that OTC deficiency in the liver leads to the build-up of ammonia, which causes chronic liver damage, and this is a major risk factor of HCC [67]. Moreover, increased liver fibrosis has been observed in heterologous OTC-KO mice [68]. Additionally, OTC overexpression has been shown to inhibit HCC cell proliferation [69]. Therefore, low OTC expression may enable tumor cells to increase ammonia accumulation, representing a loss of function of the tumor-specific metabolism of OTC. Gu et al. [66] reported that ASGR1 overexpression reduced hepatoma cell migration and invasion by interacting with LASS2. Here, we found that the expression of a serum glycoprotein homeostasis regulator, ASGR1, was downregulated in HNF4A KD cells, suggesting that HNF4A positively regulates ASGR1 expression in HCC cells. Therefore, our result is consistent with the previous report and suggests the role of ASGR1 as a tumor suppressor in HCC [70]. Furthermore, it has been reported that IYD overexpression suppressed Huh7 cell growth by inhibiting glycolysis in HCC cells [71]. Therefore, the downregulation of IYD in HNF4A KD cells is considered as a key driver in HCC malignancy, especially when both HNF1A and HNF4A have loss-of-function activity. However, the contribution of IYD in relation to HNF-family genes to tumorigenesis in the liver has not been investigated yet, and further studies are needed. HPR, PKLR and PLAU have been reported to be overexpressed in breast cancers and esophageal squamous cell carcinoma [72-74]. Conversely, our study showed the downregulation of those genes. It is possible that these genes might be tissue specific, and their downregulation may promote the transition from liver damage to hepatocarcinogenesis and enhance HCC progression in the presence of the loss of function of HNF4A/1A in HCC, but further studies are needed to validate this hypothesis.
>In addition, authors have used some exaggerating words when referred to their findings. Although your findings are interesting there could be difficulties in receiving similar outcomes by following your experimental plan in other labs or clinics.
Thank you for your comments and we have newly used milder description such as “it is possible that’’ or “our results suggest that” and removed strong words such as ‘”our result indicates” etc. Moreover, we have requested MDPI proofreading service to check our English is properly used.
>Acronyms should be spelled first, otherwise please provide a list with all abbreviations used.
Thank you for your suggestion. We have added a list for all the abbreviations used in the manuscript.
>The introduction chapter is disorganised. Please more correctly elaborate and add references to each statement. Some parts, e.g regarding third most common cancer-related death, or regarding risk factors are incorrect. So please amend.
Thank you for your comments and suggestion. We have improved our Introduction and added references to each statement including the sentence that you pointed out. As you pointed out, it is not correct to describe liver cancer as the third causative of death in cancer therefore we modified this to ‘’one of the leading causes of cancer dependent death” and add several references to support our statement. Additionally, according to our literature verification, gender, type2 diabetes, alfatoxins, Cirrhosis etc can be also considered as liver cancer risks and smoking, alcohol consumption, HBV/HCV infections can be included as major risks but not limited to them. Therefore, we modified the sentence to “The common risk factors for liver cancer development contains smoking, alcohol consumption, hepatitis B and C, and other metabolic disease”. We have added a reference to support our statement. Thank you very much for your clarification and we hope our description fulfills your criteria this time. Again, thank you for your notifications and it was quite helpful to get your advice on our manuscript.
>Please provide rs code(s) for V259F mutation and other listed throughout the manuscript
The rsID number (code) is normally used to identify a specific single nucleotide polymorphism and we have studied on somatic mutations found by ICGC consortia. Therefore, we added the mutation ID in the Table1. Thank you for your comment on this point.